# Towards Automatic Concept-based Explanations

**Amirata Ghorbani**\*
Stanford University
amiratag@stanford.edu

**James Wexler**
Google Brain
jwexler@google.com

**James Zou**
Stanford University
jamesz@stanford.edu

**Been Kim**
Google Brain
beenkim@google.com

## Abstract

Interpretability has become an important topic of research as more machine learning (ML) models are deployed and widely used to make important decisions. Most of the current explanation methods provide explanations through feature importance scores, which identify features that are important for each individual input. However, how to systematically summarize and interpret such per sample feature importance scores itself is challenging. In this work, we propose principles and desiderata for *concept* based explanation, which goes beyond per-sample features to identify higher level human-understandable concepts that apply across the entire dataset. We develop a new algorithm, ACE, to automatically extract visual concepts. Our systematic experiments demonstrate that ACE discovers concepts that are human-meaningful, coherent and important for the neural network's predictions.

## 1 Introduction

As machine learning (ML) becomes widely used in applications ranging from medicine [17] to commerce [38], gaining insights into ML models' predictions has become an important topic of study, and in some cases a legal requirement [16]. The industry is also recognizing explainability as one of the main components of responsible use of ML [1]; not just a nice-to-have component but a must-have one.

Most of the recent literature on ML explanation methods has revolved around deep learning models. Methods that are focused on providing explanations of ML models follow a common procedure: For each input to the model, they alter individual features (pixels, super-pixels, word-vectors, etc) either in the form of removal (zero-out, blur, shuffle, etc) [29, 5] or perturbation [35, 34] to approximate the importance of each feature for model's prediction. These "feature-based" explanations suffer from several drawbacks. There has been a line of research focused on showing that these methods are not as reliable [14, 3, 15]. For examples, Kindermans *et al.*discussed vulnerability even to simple shifts in the input [21] while Ghorbani *et al.*designed adversarial perturbations against these methods. A more important concern, however, is that human experiments show that these methods are susceptible to human confirmation biases [20], and also showing that these methods do not increase human understanding of the model and human trust in the model [28, 20]. For example, Kim *et al.* [20] showed that given identical feature-based explanations, human subjects confidently find evidence for completely contradicting conclusions.

As a consequence, a recent line of research has focused on providing explanations in the form of high-level human "concepts" [46, 20]. Instead of assigning importance to individual features or pixels, the

output of the method reveals the important concepts. For examples, the wheel and the police logo are important concepts for detecting police vans. These methods come with their own drawbacks. Rather than pointing to the important concepts, they respond to the user's queries about concepts. That is, for each concept's importance to query, a human has to provide hand-labeled examples of that concept. While these methods are useful when the user knows the set of well-defined concepts and has the resources to provide examples, a major problem is that the space of possible concepts to query can first of all, be unlimited, or in some settings even be unclear. Another important drawback is that they rely on human bias in the explanation process; humans might fail to choose the right concepts to query. Because these previous methods can only test concepts that are already labeled and identified by humans, their discovery power is severely limited.

**Our contribution** We lay out general principles that a concept-based explanation of ML should satisfy. Then we develop a systemic framework to automatically identify higher-level concepts which are meaningful to humans and are important for the ML model. Our novel method, Automated Concept-based Explanation (ACE), works by aggregating related local image segments across diverse data. We apply an efficient implementation of our method to a widely-used object recognition model. Quantitative human experiments and evaluations demonstrate that ACE satisfies the principles of concept-based explanation and provide interesting insights into the ML model.[2]

## 2 Concept-based Explanation Desiderata

Our goal is to explain a machine learning model's decision making via units that are more understandable to humans than individual features, pixels, characters, and so forth. Following the literature [46, 20], throughout this work, we refer to these units as concepts. A precise definition of a concept is not easy [13]. Instead, we lay out the desired properties that a concept-based explanation of a machine learning model should satisfy to be understandable by humans.

1. *Meaningfulness* An example of a concept is semantically meaningful on its own. In the case of image data, for instance, individual pixels may not satisfy this property while a group of pixels (an image segment) containing a texture concept or an object part concept is meaningful. Meaningfulness should also correspond to different individuals associating similar meanings to the concept.

2. *Coherency* Examples of a concept should be perceptually similar to each other while being different from examples of other concepts. Examples of "black and white striped" concept are all similar in having black and white stripes.

3. *Importance* A concept is "important" for the prediction of a class if its presence is necessary for the true prediction of samples in that class. In the case of image data, for instance, the object which presence is being predicted is necessary while the background color is not.

We do not claim these properties to be a complete set of desiderata, however, we believe that this is a good starting point towards concept-based explanations.

## 3 Methods

An explanation algorithm has typically three main components: A trained classification model, a set of test data points from the same classification task, and a importance computation procedure that assigns importance to features, pixels, concepts, and so forth. The method either explains an individual data point's prediction (local explanation), or an entire model, class or sets of examples (global explanation). One example of a local explanation method is the family of saliency map methods [33, 34, 35]. Each pixel in every image is assigned an importance score for the correct prediction of that image typically by using the gradient of prediction with respect to each pixel. TCAV [20] is an example of a global method. For each class, it determines how important a given concept is for predicting that class.

In what follows, we present ACE . ACE is a global explanation method that explains an entire class in a trained classifier without the need for human supervision.

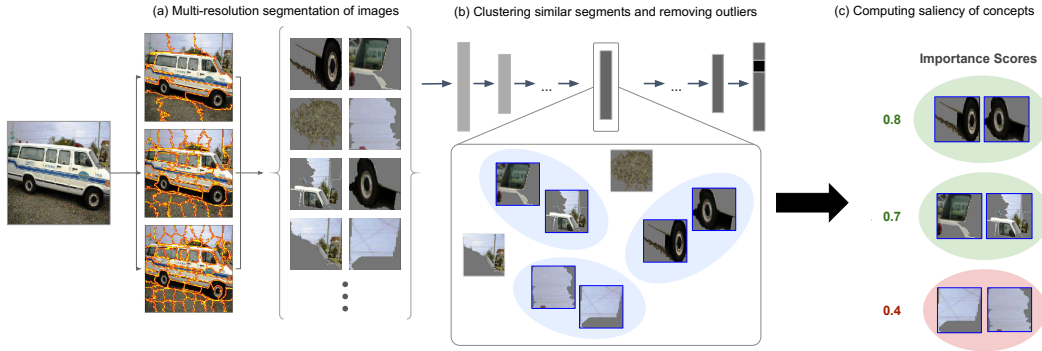

Figure 1: **ACE algorithm** (a) A set of images from the same class is given. Each image is segmented with multiple resolutions resulting in a pool of segments all coming from the same class. (b) The activation space of one bottleneck layer of a state-of-the-art CNN classifier is used as a similarity space. After resizing each segment to the standard input size of the model, similar segments are clustered in the activation space and outliers are removed to increase coherency of clusters. (d) For each concept, its TCAV importance score is computed given its examples segments.

**Automated concept-based explanations step-by-step** ACE takes a trained classifier and a set of images of a class as input. It then extracts concepts present in that class and returns each concept's importance. In image data, concepts are present in the form of groups of pixels (segments). To extract all concepts of a class, the first step of ACE (Fig 1(a) starts with segmentation of the given class images. To capture the complete hierarchy of concepts from simple fine-grained ones like textures and colors to more complex and coarse-grained ones such as parts and objects, each image is segmented with multiple resolutions. In our experiments, we used three different levels of resolution to capture three levels of texture, object parts, and objects. As discussed in Section 4, three levels of segmentation is enough to achieve the goal.

The second step of ACE (Fig 1(b)) groups similar segments as examples of the same concept. To measure the similarity of segments, we use the result of previous work [45] showing that in state-of-the art convolutional neural networks (CNNs) trained on large-scale data sets like ImageNet [32], the euclidean distance in the activation space of final layers is an effective perceptual similarity metric. Each segment is then passed through the CNN to be mapped to the activation space. Similar to the argument made by Dabkowski & Gal [8], as most image classifiers accept images of a standard size while the segments have arbitrary size, we resize the segment to the required size disregarding aspect ratio. As the results in Section 4 suggest, this works fine in practice but it should be mentioned that the proposed similarity measure works the best with classifiers robust to scale and aspect ratio. After the mapping is performed, using the euclidean distance between segments, we cluster similar segments as examples of the same concept. To preserve concept coherency, outlier segments of each cluster that have low similarity to cluster's segments are removed (Fig. 1(b)).

The last step of ACE (Fig 1(c)) includes returning important concepts from the set of extracted concepts in previous steps. TCAV [20] concept-based importance score is used in this work (Fig. 1(c)), though any other concept-importance score could be used.

**How ACE is designed to achieve the three desiderata** The first of the desiderata requires the returned concepts to be clean of meaningless examples (segments). To perfectly satisfy meaningfulness, the first step of ACE can be replaced by a human subject going over all the given images and extracting only meaningful segments. To automate this procedure, a long line of research has focused on semantic segmentation algorithms [25, 23, 27, 30], that is, to segment an image so that every pixel is assigned to a meaningful class. State-of-the art semantic segmentation methods use deep neural networks which imposes higher computational cost. Most of these methods are also unable to perform segmentation with different resolutions. To tackle these issues, ACE uses simple and fast super-pixel segmentation methods which have been widely used in the hierarchical segmentation literature [43]. These methods could be applied with any desired level of resolution with low computational cost

at the cost of suffering from lower segmentation quality, that is, returning segments that either are meaningless or capture numerous textures, objects, etc instead of isolating one meaningful concept.

To have perfect meaningfulness and coherency, we can replace the second step with a human subject to go over all the segments, clusters similar segments as concepts, and remove meaningless or dissimilar segments. The second step of ACE aims to automate the same procedure. It replaces a human subject as a perceptual similarity metric with an ImageNet-trained CNN. It then clusters similar segments and removes outliers. The outlier removal step is necessary to make every cluster of segments clean of meaningless or dissimilar segments. The idea is that if a segment is dissimilar to segments in a cluster, it is either a random and meaningless segment or if it is meaningful, it belongs to a different concept; a concept that has appeared a few times in the class images and therefore its segments are not numerous enough to form a cluster. For example, asphalt texture segments are present in almost every police van image and therefore are expected to form a coherent cluster while segments of grass texture that are present in only one police van image form an unrelated concept to the class and are to be removed.

ACE utilizes the TCAV score as a concept's importance metric. The intuition behind the TCAV score is to approximate the average positive effect of a concept on predicting the class and is generally applied to deep neural network classifiers. Given examples of a concept, TCAV score [20] is the fraction of class images for which the prediction score increases if the representation of those images in the activation space are perturbed in the general direction of representation of concept examples in the same activation space (with the use of directional derivatives). Details are described in the original work [20].

It is evident that satisfying the desiderata through ACE is limited to the performance of the segmentation method, the clustering and outlier removal method, and above all the reliability of using CNNs as a similarity metric. The results and human experiments in the next section verify the effectiveness of this method.

## 4   Experiments and Results

As an experimental example, we use ACE to interpret the widely-used Inception-V3 model [36] trained on ILSVRC2012 data set (ImageNet) [32]. We select a subset of 100 classes out of the 1000 classes in the data set to apply ACE . As shown in the original TCAV paper [20], this importance score performs well given a small number of examples for each concept (10 to 20). In our experiments on ImageNet classes, 50 images was sufficient to extract enough examples of concepts; possibly because the concepts are frequently present in these images. The segmentation step is performed using SLIC [2] due to its speed and performance (after examining several super-pixel methods [10, 26, 41]) with three resolutions of 15, 50, and 80 segments for each image. For our similarity metric, we examined the euclidean distance in several layers of the ImageNet trained Inception-V3 architecture and chose the "mixed_8" layer. As previously shown [20], earlier layers are better at similarity of textures and colors while latter ones are better for object and the "mixed_8" layer yields the best trade-off. K-Means clustering is performed and outliers are removed using euclidean distance to the cluster centers. More implementation details are provided in Appendix A.

**Examples of ACE algorithm**   We apply ACE to 100 randomly selected ImageNet classes. Fig. 2 depicts the outputs for three classes. For each class, we show the four most important concepts via three randomly selected examples (each example is shown above the original image it was segmented from). The figure suggests that ACE considers concepts of several levels of complexity. From Lionfish spines and its skin texture to a car wheel or window. More examples are shown in Appendix E.

**Human experiments**   To verify the coherency of concepts, following the explainability literature [7], we designed an intruder detection experiment. At each question, a subject is asked to identify one image out of six that is conceptually different from the rest. We created a questionnaire of 34 questions, such as the one shown in Fig. 3. Among 34 randomly ordered questions, 15 of them include using the output concepts of ACE and other 15 questions using human-labeled concepts from Broaden dataset [4]. The first four questions were used for training the participants and were discarded. On average, 30 participants answered the hand-labeled dataset 97% (14.6/15) ($\pm 0.7$) correctly, while discovered concepts were answered 99% (14.9/15) ($\pm 0.3$) correctly. This experiment

# Lionfish    Police Van    Basketball

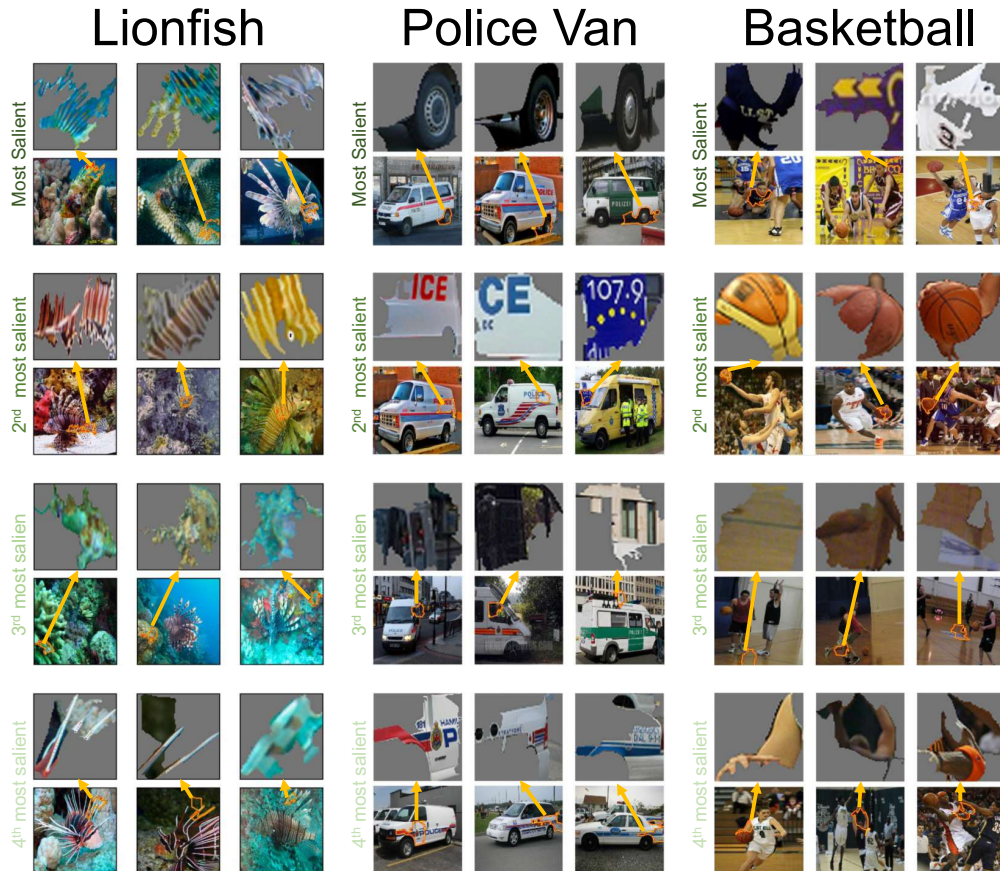

Figure 2: **The output of ACE for three ImageNet classes.** Here we depict three randomly selected examples of the top-4 important concepts of each class (each example is shown above the original image it was segmented from). Using this result, for instance, we could see that the network classifies police vans using the van's tire and the police logo.

confirms that while a discovered concept is only a set of image segments, ACE outputs segments that are *coherent*.

In our second experiment, we test how meaningful the concepts are to humans. We asked 30 participants to perform two tasks: As a baseline test of meaningfulness, first we ask them to choose the more meaningful of two options. One being four segments of the same concept (along with the image they were segmented from) and the other being four random segments of images in the same class. the right option was chosen $95.6\%$ $(14.3/15)(\pm1.0)$. To further query the meaningfulness of the concepts, participants were asked to describe their chosen option with one word. As a result, for each question, a set of words (e.g. `bike`, `wheel`, `motorbike`) are provided and we tally how many individuals use the same word to describe each set of image. For examples, for the question in Fig. 3, 19 users used the word `human` or `person` and 8 users used `face` or `head`. For all of the questions, on average, $56\%$ of participants described it with the most frequent word and its synonyms ($77\%$ of descriptions were from the two most frequent words). This suggests that, first of all ACE discovers concepts with high precision. Secondly, the discovered concepts have consistent semantic/verbal meanings across individuals. The questionnaire had 19 questions and the first 4 were used as training and were discarded.

**Examining the importance of important concepts**    To confirm the importance scores given by TCAV, we extend the two importance measures defined for pixel importance scores in the literature [8] to the case of concepts. Smallest sufficient concepts (SSC) which looks for the smallest set of concepts that are enough for predicting the target class. Smallest destroying concepts (SDC) which looks for the smallest set of concepts removing which will cause incorrect prediction. Note that

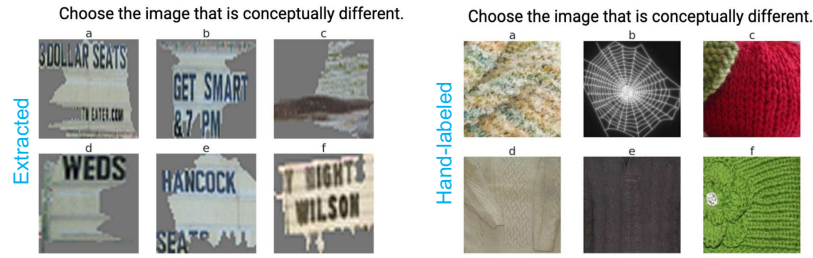

Experiment 1: Identifyig intruder concept

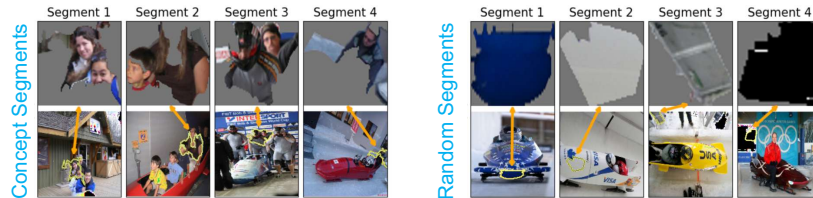

Experiment 2: Identifying the meaning of concept

Figure 3: **Human subject experiments questionnaires.** (Texts in blue are not part of the questionnaire) (a) 30 human subjects were asked to identify one image out of six that is conceptually different from the rest. For comparison, each question is either a set of extracted or hand-labeled concepts. On average, participants answer the hand-labeled dataset 97% (14.6/15, ±0.7) correctly, while discovered concepts were answered 99% (14.9/15, ±0.3) correctly. (b) 30 human subjects were asked to identify a set of image segments belonging to a concept versus a random set of segments and then to assign a word to the selected concept. On average, 55% of participants used the most frequent word and its synonyms for each question and 77% of the answers were one of top-two frequent words.

although these importance scores are defined and used for local pixel-based explanations in [8] (explaining one data point), the main idea can still be used to evaluate our global concept-based explanation (explaining a class).

To examine ACE with these two measures, we use 1000 randomly selected ImageNet validation images from the same 100 classes. Each image is segmented with multiple resolutions similar to ACE . Using the same similarity metric in ACE , each resulting segment is assigned to a concept using its the examples of a concept with least similarity distance concept's examples. Fig. 4 shows the prediction accuracy on these examples as we add and remove important concepts.

**Insights into the model through ACE**    To begin with, some interesting correlations are revealed. For many classes, the concepts with high importance follow human intuition, e.g. the "Police" characters on a police car are important for detecting a police van while the asphalt on the ground is not important. Fig. 5(a) shows more examples of this kind. On the other side, there are examples where the correlations in the real world are transformed into model's prediction behavior. For instance, the most important concept for predicting basketball images is the players' jerseys rather than the ball itself. It turns out that most of the ImageNet basketball images contain jerseys in the image (We inspected 50 training images and there was a jersey in 48 of them). Similar examples are shown in Fig. 5(b). A third category of results is shown in Fig. 5(c). In some cases, when the object's structure is complex, parts of the object as separate concepts have their own importance and some parts are more important than others. The example of carousel is shown: lights, poles, and seats. It is interesting to learn that the lights are more important than seats.

A natural follow-up question is whether the mere existence of a important concepts is enough for prediction without having the structural properties; e.g. an image of just black and white zebra stripes is predicted as zebra. For each class, we randomly place examples of the four most important

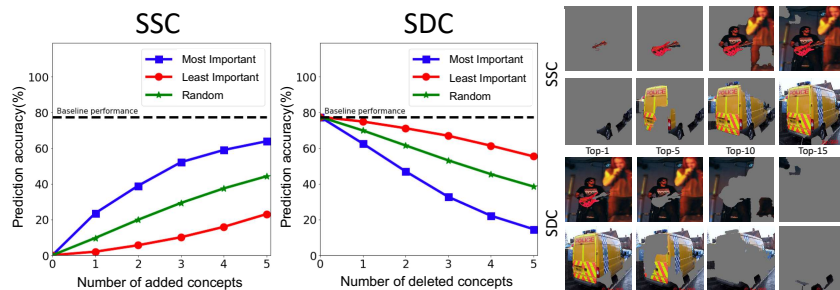

Figure 4: **Importance** For 1000 randomly sampled images in the ImageNet validation set, we start removing/adding concepts from the most important. As it is shown, the top-5 concepts is enough to reach within $80\%$ of the original accuracy and removing the top-5 concepts results in misclassification of more than $80\%$ of samples that are classified correctly. For comparison, we also plot the effect of adding/removing concepts with random order and with reverse importance order.

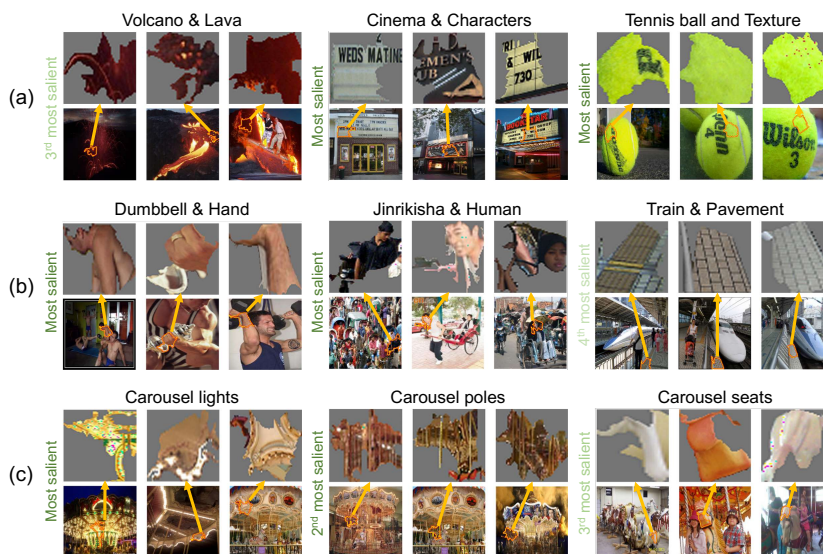

Figure 5: **Insights into the model** The text above each image describes its original class and our subjective interpretation of the extracted concept; e.g. "Volcano" class and "Lava" concepts. (a) Intuitive correlations. (b) Unintuitive correlations (c) Different parts of an object as separate but important concepts

concepts on a blank image. (100 images for each class) Fig. 6 depicts examples of these randomly "stitched" images with their predicted class. For 20 classes (zebra, liner, etc), more than $80\%$ of images were classified correctly. For more than half of the classes, above $40\%$ of the images were classified correctly (note that random chance is $0.001\%$). This result aligns with similar findings [6, 12] of surprising effectiveness of Bag-of-local-Features and CNNs bias towards texture and shows that our extracted concepts are important enough to be *sufficient* for the ML model. Examples are discussed in Appendix C.

## 5 Related Work

This work is focused on post-training explanation methods - explaining an already trained model instead of building an inherently explainable model [42, 19, 40]. Most common post-training explanation methods provide explanations by estimating the importance of each input feature (covariates, pixels, etc) or training sample for the prediction of a particular data point [33, 34, 44, 22] and are designed to explain the prediction on individual data points. While this is useful when only specific data points matter, these methods have been shown to come with many limitations, both methodologically

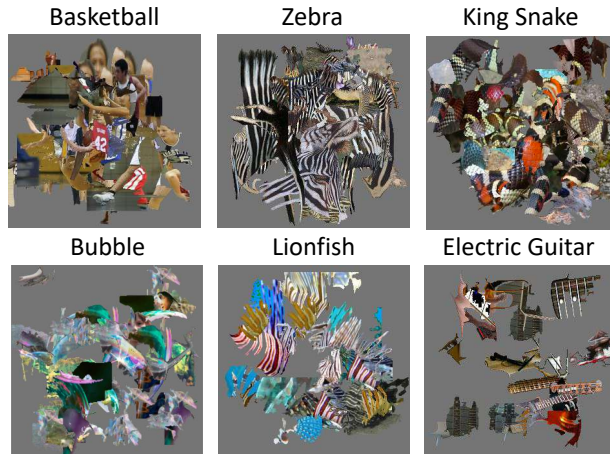

Figure 6: **Stitching important concepts** We test what would the classifier see if we randomly stitch important concepts. We discover that for a number classes this results in predicting the image to be a member of that class. For instance, basketball jerseys, zebra skin, lionfish, and king snake patterns all seem to be enough for the Inception-V3 network to classify them as images of their class.

and fundamentally. [21, 18, 14] For example, [18] showed that some input feature-based explanations are qualitatively and quantitatively similar for a trained model (i.e., making superhuman performance prediction) and a randomized model (i.e., making random predictions). Other work proved that some of these methods are in fact trying to reconstruct the input image, rather than estimating pixels' importance for prediction [39]. In addition, it's been shown that these explanations are susceptible to humans' confirmation biases [20]. Using input features as explanations also introduces challenges in scaling this method to high dimensional datasets (e.g., health records). Humans typically reason in higher abstracted concepts [31] than a particular input feature (e.g., lab results, a particular hospital visit). A recently developed method uses high-level concepts, instead of input features. TCAV [20] produces estimates of how important that a concept was for the prediction and IBD [46] decomposes the prediction of one image into human-interpretable conceptual components. Both methods require humans to provide examples of concepts. Our work introduces an explanation method that explains each class in the network using concepts that are present in the images of that class while removing the need for humans to label examples of those concepts. [37]

## 6    Discussion

We note a couple of limitations of our method. The experiments are performed on image data, as automatically grouping features into meaningful units is simple for this case. The general idea of providing concept-based explanations applies to to other data types such as texts, and this would be an interesting direction of future work. An interesting direction of future work here is to apply more sophisticated dictionary learning approaches, beyond clustering, on the representation space (e.g. sparse coding) which could reduce the need for image segmentation, and learn more complex concepts. Additionally, the above discussions only apply to concepts that are present in the form of groups of pixels. While this assumption gave us plenty of insight into the model, there might be more complex and abstract concepts are difficult to automatically extract. Future work includes tuning the ACE hyper-parameters (multi-resolution segmentation, etc) for each class separately. This may better capture the inherent granularity of objects; for example, scenes in nature may contain a smaller number of concepts compared to scenes in a city.

In conclusion, we introduces ACE , a post-training explanation method that automatically groups input features into high-level concepts; *meaningful* concepts that appear as *coherent* examples and are *important* for correct prediction of the images they are present in. We verified the meaningfulness and coherency through human experiments and further validated that they indeed carry salient signals for prediction. The discovered concepts reveal insights into potentially surprising correlations that the model has learned. Such insights may help to promote safer use of this powerful tool, machine learning.

**Acknowledgement**    A.G. is supported by a Stanford Graduate Fellowship (Robert Bosch Fellow). J.Z. is supported by NSF CCF 1763191, NIH R21 MD012867-01, NIH P30AG059307, and grants from the Silicon Valley Foundation and the Chan-Zuckerberg Initiative.

## Footnotes

\*Work done while interning at Google Brain.

[2] Implementation available: `https://github.com/amiratag/ACE`

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
