[Supplementary Material · ACE_supp.pdf]

# A   More Implementation Details

We selected a random subset of 100 classes in ImageNet dataset and chose a random set of 50 images in the training set of each class to be our "concept-discover" images. For each class, we performed SLIC super-pixel segmentation on the discovery-images. Each of the images was segmented into 15, 50 and 80 segments. Each segment was then resized to the original input size of Inception-V3 network (by padding with gray scale value of 117.5 which is the default zero value for Inception-V3 network). We then pass all the segments of a class through the Inception-V3 network to get their "mixed_8" layer representation.

The next step is to cluster the segments that belong to one concept together (*coherent* examples) while removing the meaningless segments. We tested several clustering methods including K-means [24], Affinity Propagation [11], and DBSCAN [9]. When Affinity Propagation was used, typically a large number of clusters (30-70) were produced, which was then simplified by another hierarchical clustering step. Interstingly, the best results were acquired as follows: We first perform K-Means clustering with $K = 25$. After performing the K-Means, for each cluster, we keep only the $n = 40$ segments that have the smallest $\ell_2$ distance from the cluster center and remove the rest. We then remove three types of clusters: 1) Infrequent Clusters that have segments only coming from one or a few number of discovery-images. The problem with these clusters is that the concept they represent is not a common concept for the target-class. One example is that having many segments of the same grass type that appears in one image. These segments tend to form a cluster due to similarity but don't represent a frequent concept. 2) Unpopular clusters that have very few members. To have a trade-off, we keep three groups of clusters: a) high frequency (segments come from more than half of discovery images) b) medium frequency with medium popularity (more than one-quarter of discovery images and the cluster size is larger than the number of discovery-images) and c) high popularity (cluster size is larger than twice the number of discovery images.)

# B   ACE Considers Simple to Complex Concepts

The multi-resolution segmentation step of ACE naturally returns segments that contain simple concepts such as color or texture and more complex concepts, such as parts of body or objects. Among those segments, ACE successfully identifies concepts with similar level of abstract-ness with similar semantic meaning (as verified via human experiment). Supp. Fig. 1 shows some examples of the discovered concepts. Note that each segment is re-sized for display.

Supplementary Figure 1: **Examples of discovered concepts.** A wide range of concepts like blue color, asphalt texture, car window, and human face are detected through the algorithm. Multi-resolution segmentation helped discovering concepts with varying sizes. For example, two car windows with different sizes (one twice as big as the other) were identified as the same concept.

# C   Drawbacks of ACE

The first drawback of ACE is its susceptibility for returning either meaningless or non-coherent concepts due to the segmentation errors, clustering errors, or errors of the similarity metric. While rare, there are concepts that are less subjectively less coherent. This may be due to limitations of our method or because things that are similar to the neural network are not similar to humans. However, the incoherent concepts were never in top-5 most important concepts among the 100 classes used for experiments. Another potential problem is the possibility of returning several concepts that are subjectively duplicates. For example, in Supp. Fig. 2(b), three different ocean surfaces (wavy, calm, and shiny) are discovered separately and all of them have similarly high TCAV scores. Future work remains to see whether this is because the network represents these ocean surfaces differently, or whether we can further combine these concepts into one 'ocean' concept.

Supplementary Figure 2:  (a) Semantically inconsistent concepts achieve low or no TCAV scores (b) Seemingly duplicated concepts (to humans) may be discovered

# D  Stitching Concepts

Supplementary Figure 3: Examples of stitched images classified correctly by the Inception-V3 network.

Supplementary Figure 4: Examples of stitched images classified correctly by the Inception-V3 network.

# E    More Examples of ACE

We show the results for 12 ImageNet classes. For each class, four of the top-5 important concepts are shown.

Supplementary Figure 5: More examples of ACE.

# Gorilla          Park Bench          Casette

Supplementary Figure 6: More examples of ACE.

Shopping Cart    Norfolk Terrier    Jeep

Supplementary Figure 7: More examples of ACE.

# Boxer  School Bus  Lotion

Supplementary Figure 8: More examples of ACE.