[Reviews · NeurIPS 2019]

Reviewer 1



This paper examines the use of concept-based explanations rather than feature-base explanations. The algorithm in the paper, ACE, segments images, clusters similar segmentations, and then provides a list of explanations (salient concepts) by using TCAV. The paper had very good motivation. I think that the authors could also motivate the use of concpet-level explanations to mimic commonsense. For example, in the police van example, the individual features are not as important as the police logo and the general shape of the vehicle. Even if we are new countries and places with different polic vans, we can still abstract out the higher level concepts (like shape and logo) that distinguish the vehicle as a police type. The concept-based explanation desirata is a nice, concise section outlining the standards and evaluations for concept-based explanations. A small point, but the authors may want to distiguish the difference between a saliency property and a saliency map. When first reading the paper, I was confused whether the properties in the desiderata were types of methods or properties of the output. The methods secion provides a nice overview and definition of explanatory algorithm. I found the description and algorithm of ACE to be clear and straightforward. The authors state that the second step of ACE (the clustering of similar segmentations) can be replaced with a human subject, and I was wondering if this was tested or an effective augmentaion. That could be a nice sanity check on top of some of these concepts. In addition, the authors motivate the use of TCAV for a concept score. Although, they say "any other concept-importance saliency score could be used," I was wondering (1) why they choose to use TCAV rather than another salience score (e.g. Network Dissection) and what other scores could be used (or not) and why. I was also wondering about the last point about the reliability of CNNs as a similarity metric. I was left wondering if this method could point out adversarial examples. There were a few details in the experiment and results section that could be explained further. For example, why was using 50 images sufficient for extracting the related concepts? The authors state that this is because "the related concepts of a class tend to be present frequently in the class images," but I was wondering if this is dependent on the dataset. Similarly, I was wondering why the authos applied ACE to 100 ImageNet classes. Although I was delighted to see the results in the Appendix. I think the user study could use a bit more quantification, though. While I was convinced with the results, I was wondering how similar descriptions were between participants. Although the authors stated that "77% of descriptions were from the two most frequent words on average," I wasn't exactly sure what they meant across the experiment. The example was motivating, but I think a similarity score may strenghten the argument. In summary, I thought this was a nice paper that extended TCAV and promoted the use of concept-based explanations. Although I found the method and results to be sound, I was left wondering about future work and applications of such a method. I think the idea of using this to harden existing DNNs is a good one, but I was left wondering how that would be done.

Reviewer 2



The paper is clear, well written and addresses an important problem in ML research, specially for black-box models such DNNs. The authors propose a set of desiderata for concept-based explanation models and novel method to produce global explanations for vision DNNs. The proposed method accomplish all the criteria by replicating the manual process of finding relevant concepts byt replace humans with automated methods. The techniques used to implement the method are not novel but they were combined in a novel and relevant way. The works was thoroughly evaluated with appropriate methodologies, including an intrusion task to measure coherency, SSC and SDC to measure saliency and inspection of the learned model.

Reviewer 3



Originality: The paper is clearly linked to dictionary learning, unsupervised feature extraction, and also with the original LIME paper (where the interpretable features aka concepts are designed by hand). Regardless, the paper seems completely original. Quality: I really like the idea of providing explanations using higher level concepts, and also the fact that the authors did carry out experiments with human subjects. The proposed method is very, very simple (it is basically segmentation + clustering w.r.t. the inception distance + TCAV afterwards), but this is not necessarily an issue. I also really like Figure 6, which should be given more prominence, I think. Still, I do not really like the desiderata, which feel ill-defined, inconsistent, and overall vacuous: - Meaningfulness to me sounds like it is referring to concepts with a meaning---where both "concept" and "meaning" are not well defined. - Coherency, or actually "perceptual similarity": it seems to be much lower level than meaningfulness. For instance, cars definitely represent a meaningful concept, but they can be perceptually very different to each other. The same goes for most concrete concepts (person, chair, phone, ...). After reading meaningfulness and coherency, I don't know if the authors would consider "zebra pattern" as a high-level concept or just a texture pattern. These two desiderata don't seem to be helping my understanding. I feel like the desiderata don't add much to the paper (if not confusion), and should probably be compressed to two / three inline sentences. As things stand now, I feel like the desiderata detract from the presentation. Also, the proposed method is very image-centric; this is probably a liftable limitation. The second thing that I don't like is that, at a higher level, the paper carries a very bad message: that debugging models using automatically extracted concepts would remove the need for human intervention (line 47). The problem is that if the extracted concepts are not good (and segmenters *can* extract garbage, despite the experiments with human subjects presented here), then it is impossible for anybody to debug the learned model. I would prefer if the message was about helping or augmenting human experts, not about replacing them. It seems both harmful and irrealistic. Clarity: The paper is reasonably well structured, but the presentation is not always clear. For instance: - lines 27-28 are unclear - line 47: "ace **removes** the need to have humans look ...": way too strong; please reword. - line 61: "meaningfulness should also [...]", I cannot parse this sentence. - lines 108-131 are extremely dense and hard to parse; please rewrite. - line 122: "replaces of" - line 150: "examined at" Most importantly, the description of the human pilot experiments are *very* hard to follow, and should be heavily revised. Significance: The paper touches upon an very important topic and proposes a useful baseline solution approach. I am confident that it would be of interest to other researchers. Figure 6 is especially interesting, and I am sure it will garner a lot of attention.

[Author Response · NeurIPS 2019]

We thank all the reviewers for their valuable feedback.

**Response to Reviewer 1**    We thank the reviewer for the helpful feedback. We agree with the reviewer about the possible confusion regarding the term "saliency", and will clarify this in the text. Thanks for the suggestion on communicating the results in Fig. 6; we will improve the visualization here. We will also correct all of the typos.

We agree with the reviewer that a good concept-based explanation method should be closely aligned with human common-sense. ACE is a step toward this goal, and it opens the door for many interesting follow up works. Regarding the choice of importance score, we chose TCAV score due to its simplicity and its compatibility with our goal. Given a few examples of a concept, TCAV returns its scalar importance value for the prediction of a target class. ACE can be combined with other global interpretation methods such as (Yeh et al. NeurIPS 2018) to further select 'good examples' to summarize the discovered concept, and this is a good direction of additional work.

We have added more details and discussions of the experiments based on the reviewer's feedback. First, the reviewer asked why we used 50 images for each class. As shown in the original TCAV paper, this importance score performs well given a few number of examples for each concept (10 to 20). In our experiments on ImageNet classes, 50 images was sufficient, possibly because the concepts are frequently present in these images. Second, the reviewer refers to the discussion of human experiment results. For each question (i.e. set of images) in the experiment, participants were asked to provide a one-word explanation of the shown concept. As a result, for each question, a set of words (e.g. `bike`, `wheel`, `motorbike`) are provided and we tally how many individuals use the same word to describe each set of image. We then find the most frequent word for each image set. Averaged across the 15 questions in the experiment, $56\%$ of the participants used the same word to describe the image set, and $77\%$ of the participants used one of the two most frequent words. Because many participants independently came up with the same word to describe a set of images, this strongly suggests that each set of images is capturing one coherent concept that's meaningful to the participants. It is also interesting to compute the similarity of individuals in answering the questionnaire. For every two participant in the experiment, on average they give the same answer to $5.8$ out of the 15 questions. We will make this clear in the revised paper. We find the suggestion of replacing humans with a second step of ACE (or in general testing humans in the context of concept-based explanations) very interesting and we believe this suggestion could be a very informative project of its own.

**Response to Reviewer 2**    We thank reviewer for the thoughtful feedback and we are happy that you liked the paper! Our ACE framework is natural and easy to implement, and it opens the door for several interesting directions for future work in concept based explanations. In the revision, we have improved the discussion and provided more details on our experiments and validations. We will also release a practitioner-friendly implementation of ACE.

**Response to Reviewer 3**    Thank you for your helpful feedback. We agree with you that the goal of our work is in augmenting (and not replacing) human experts in explanation of the models. We will further emphasize this point in the text and will delete the sentence on line 47. We will also add discussions of connections to dictionary learning; thanks for this suggestion. A very interesting direction of future work here is to apply more sophisticated dictionary learning approaches, beyond clustering, on the representation space (e.g. sparse coding). This could potentially reduce the need of image segmentation, and learn more complex concepts. As mentioned in the text, we agree with you that rescaling the image patches could introduce noise in some settings. Previous work (Dabkowski et al, NeurIPS 2017) has found that ImageNet classifiers is relatively robust to changes in aspect ratio of segments; our experiments also verified that ACE is robust to aspect scaling. This is a good point to investigate further in follow up works.

We are glad that you found Figure 6 interesting; we will emphasize it more in the text. In this paper, we focused on visual concepts in image data as a good starting point, because it is relatively easy to illustrate the concepts in this setting. It'd be very interesting to expand the method to other modalities such as text; e.g. we could investigate having a cluster made of relevant words and phrases as a unit of explanation in NLP applications. This is an interesting direction of future work. We will make the discussion of the desiderata more concise and clearer (we will also move it to the Discussion section later in the paper to improve the flow).

[Meta-Review · NeurIPS 2019]

Presenting concept-based explanations rather than typical feature-base explanations, this paper offers a new algorithm, ACE, which segments and clusters the high level features of an image and provides an explanation via TCAV. Oveall the reviewers found the method well-motivated, clear and that the work was thoroughly evaluated. Several comments and issues about the presentation were raised by the reviewers. In particular the desiderata seems to be somewhat vague.